# RETHINKING ARCHITECTURE SELECTION IN DIFFERENTIABLE NAS

**Ruochen Wang[1], Minhao Cheng[1], Xiangning Chen[1], Xiaocheng Tang[2], Cho-Jui Hsieh[1]**
[1]Department of Computer Science, UCLA, [2]DiDi AI Labs
{ruocwang, mhcheng}@ucla.edu  {xiangning, chohsieh}@cs.ucla.edu
xiaochengtang@didiglobal.com

## ABSTRACT

Differentiable Neural Architecture Search is one of the most popular Neural Architecture Search (NAS) methods for its search efficiency and simplicity, accomplished by jointly optimizing the model weight and architecture parameters in a weight-sharing supernet via gradient-based algorithms. At the end of the search phase, the operations with the largest architecture parameters will be selected to form the final architecture, with the implicit assumption that the values of architecture parameters reflect the operation strength. While much has been discussed about the supernet's optimization, the architecture selection process has received little attention. We provide empirical and theoretical analysis to show that the magnitude of architecture parameters does not necessarily indicate how much the operation contributes to the supernet's performance. We propose an alternative perturbation-based architecture selection that directly measures each operation's influence on the supernet. We re-evaluate several differentiable NAS methods with the proposed architecture selection and find that it is able to extract significantly improved architectures from the underlying supernets consistently. Furthermore, we find that several failure modes of DARTS can be greatly alleviated with the proposed selection method, indicating that much of the poor generalization observed in DARTS can be attributed to the failure of magnitude-based architecture selection rather than entirely the optimization of its supernet.

## 1 INTRODUCTION

Neural Architecture Search (NAS) has been drawing increasing attention in both academia and industry for its potential to automatize the process of discovering high-performance architectures, which have long been handcrafted. Early works on NAS deploy Evolutionary Algorithm (Stanley & Miikkulainen, 2002; Real et al., 2017; Liu et al., 2017) and Reinforcement Learning (Zoph & Le, 2017; Pham et al., 2018; Zhong et al., 2018) to guide the architecture discovery process. Recently, several one-shot methods have been proposed that significantly improve the search efficiency (Brock et al., 2018; Guo et al., 2019; Bender et al., 2018).

As a particularly popular instance of one-shot methods, DARTS (Liu et al., 2019) enables the search process to be performed with a gradient-based optimizer in an end-to-end manner. It applies continuous relaxation that transforms the categorical choice of architectures into continuous architecture parameters $\alpha$. The resulting supernet can be optimized via gradient-based methods, and the operations associated with the largest architecture parameters are selected to form the final architecture. Despite its simplicity, several works cast doubt on the effectiveness of DARTS. For example, a simple randomized search (Li & Talwalkar, 2019) outperforms the original DARTS; Zela et al. (2020) observes that DARTS degenerates to networks filled with parametric-free operations such as the skip connection or even random noise, leading to the poor performance of the selected architecture.

While the majority of previous research attributes the failure of DARTS to its supernet optimization (Zela et al., 2020; Chen & Hsieh, 2020; Chen et al., 2021), little has been discussed about the validity of another important assumption: *the value of $\alpha$ reflects the strength of the underlying operations.* In this paper, we conduct an in-depth analysis of this problem. Surprisingly, we find that in many cases, $\alpha$ does not really indicate the operation importance in a supernet. Firstly, the operation associated

with larger $\alpha$ does not necessarily result in higher validation accuracy after discretization. Secondly, as an important example, we show mathematically that the domination of skip connection observed in DARTS (i.e. $\alpha_{skip}$ becomes larger than other operations.) is in fact a reasonable outcome of the supernet's optimization but becomes problematic when we rely on $\alpha$ to select the best operation.

If $\alpha$ is not a good indicator of operation strength, how should we select the final architecture from a pretrained supernet? Our analysis indicates that the strength of each operation should be evaluated based on its contribution to the supernet performance instead. To this end, we propose an alternative perturbation-based architecture selection method. Given a pretrained supernet, the best operation on an edge is selected and discretized based on how much it perturbs the supernet accuracy; The final architecture is derived edge by edge, with fine-tuning in between so that the supernet remains converged for every operation decision. We re-evaluate several differentiable NAS methods (DARTS (Liu et al., 2019), SDARTS (Chen & Hsieh, 2020), SGAS (Li et al., 2020)) and show that the proposed selection method is able to consistently extract significantly improved architectures from the supernets than magnitude-based counterparts. Furthermore, we find that the robustness issues of DARTS can be greatly alleviated by replacing the magnitude-based selection with the proposed perturbation-based selection method.

## 2 BACKGROUND AND RELATED WORK

**Preliminaries of Differentiable Architecture Search (DARTS)**   We start by reviewing the formulation of DARTS. DARTS' search space consists of repetitions of cell-based microstructures. Every cell can be viewed as a DAG with N nodes and E edges, where each node represents a latent feature map $x^i$, and each edge is associated with an operation $o$ (e.g. *skip_connect*, *sep_conv_3x3*) from the search space $\mathcal{O}$. Continuous relaxation is then applied to this search space. Concretely, every operation on an edge is activated during the search phase, with their outputs mixed by the architecture parameter $\alpha$ to form the final mixed output of that edge $\bar{m}(x^i) = \sum_{o \in \mathcal{O}} \frac{\exp \alpha_o}{\sum_{o'} \exp \alpha_{o'}} o(x^i)$. This particular formulation allows the architecture search to be performed in a differentiable manner: DARTS jointly optimizes $\alpha$ and model weight $w$ with the following bilevel objective via alternative gradient updates:

$$\min_{\alpha} \quad \mathcal{L}_{val}(w^*, \alpha) \quad \text{s.t.} \quad w^* = \arg\min_{w} \mathcal{L}_{train}(w, \alpha). \tag{1}$$

We refer to the continuous relaxed network used in the search phase as the **supernet** of DARTS. At the end of the search phase, the operation associated with the largest $\alpha_o$ on each edge will be selected from the supernet to form the final architecture.

**Failure mode analysis of DARTS**   Several works cast doubt on the robustness of DARTS. Zela et al. (2020) tests DARTS on four different search spaces and observes significantly degenerated performance. They empirically find that the selected architectures perform poorly when DARTS' supernet falls into high curvature areas of validation loss (captured by large dominant eigenvalues of the Hessian $\nabla^2_{\alpha,\alpha} \mathcal{L}_{val}(w, \alpha)$). While Zela et al. (2020) relates this problem to the failure of supernet training in DARTS, we examine it from the architecture selection aspects of DARTS, and show that much of DARTS' robustness issue can be alleviated by a better architecture selection method.

**Progressive search space shrinking**   There is a line of research on NAS that focuses on reducing the search cost and aligning the model sizes of the search and evaluation phases via progressive search space shrinking (Liu et al., 2018; Li et al., 2019; Chen et al., 2021; Li et al., 2020). The general scheme of these methods is to prune out weak operations and edges sequentially during the search phase, based on the magnitude of $\alpha$ following DARTS. Our method is orthogonal to them in this respect, since we select operations based on how much it contributes to the supernet's performance rather than the $\alpha$ value. Although we also discretize edges greedily and fine-tune the network in between, the purpose is to let the supernet recover from the loss of accuracy after discretization to accurately evaluate operation strength on the next edge, rather than to reduce the search cost.

# 3 THE PITFALL OF MAGNITUDE-BASED ARCHITECTURE SELECTION IN DARTS

In this section, we put forward the opinion that the architecture parameter $\alpha$ does not necessarily represent the strength of the underlying operation in general, backed by both empirical and theoretical evidence. As an important example, we mathematically justify that the skip connection domination phenomenon observed in DARTS is reasonable by itself, and becomes problematic when combined with the magnitude-based architecture selection.

## 3.1 $\alpha$ MAY NOT REPRESENT THE OPERATION STRENGTH

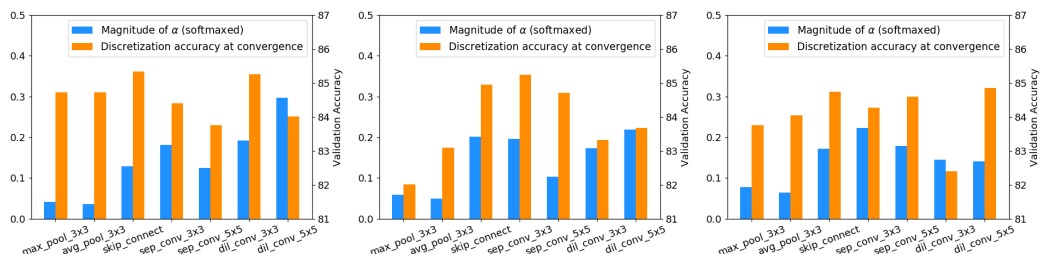

Figure 1: $\alpha$ vs discretization accuracy at convergence of all operations on 3 randomly selected edges from a pretrained DARTS supernet (one subplot per edge). The magnitude of $\alpha$ for each operation does not necessarily agree with its relative discretization accuracy at convergence.

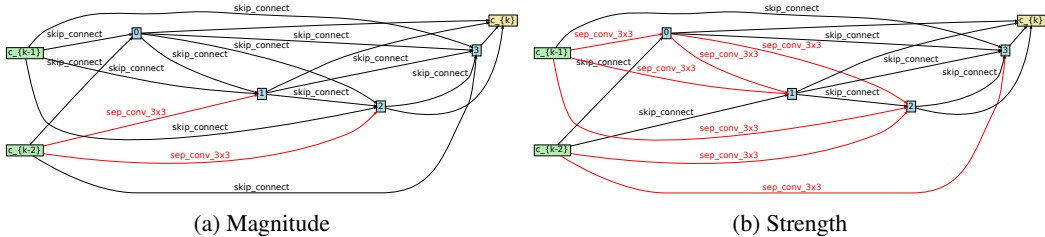

        (a) Magnitude                                        (b) Strength

Figure 2: Operation strength on each edge of S2 *(skip_connect, sep_conv_3x3)*. (a). Operations associated with the largest $\alpha$. (b). Operations that result in the highest discretization validation accuracy at convergence. Parameterized operations are marked red.

Following DARTS, existing differentiable NAS methods use the value of architecture parameters $\alpha$ to select the final architecture from the supernet, with the implicit assumption that $\alpha$ represents the strength of the underlying operations. In this section, we study the validity of this assumption in detail.

Consider one edge on a pretrained supernet; the strength of an operation on the edge can be naturally defined as the supernet accuracy after we discretize to this operation and fine-tune the remaining network until it converges again; we refer to this as "discretization accuracy at convergence" for short. The operation that achieves the best discretization accuracy at convergence can be considered as the best operation for the given edge. Figure 1 shows the comparison of $\alpha$ (blue) and operation strength (orange) of randomly select edges on DARTS supernet. As we can see, the magnitude of $\alpha$ for each operation does not necessarily agree with their relative strength measured by discretization accuracy at convergence. Moreover, operations assigned with small $\alpha$s are sometimes strong ones that lead to high discretization accuracy at convergence. To further verify the mismatch, we investigate the operation strength on search space S2, where DARTS fails dramatically due to excessive skip connections (Zela et al., 2020). S2 is a variant of DARTS search space that only contains two operations per edge *(skip_connect, sep_conv_3x3)*. Figure 2 shows the selected operations based on $\alpha$ (left) and operation strength (right) on all edges on S2. From Figure 2a, we can see that $\alpha_{skip\_connect} > \alpha_{sep\_conv\_3x3}$ on 12 of 14 edges. Consequently, the derived child architecture will lack representation ability and perform poorly due to too many skip connections. However, as shown

in Figure 2b, the supernet benefits more from discretizing to *sep_conv_3x3* than *skip_connect* on half of the edges.

## 3.2 A CASE STUDY: SKIP CONNECTION

Several works point out that DARTS tends to assign large $\alpha$ to skip connections, resulting in shallow architectures with poor generability (Zela et al., 2020; Liang et al., 2019; Bi et al., 2019). This "skip connection domination" issue is generally attributed to the failure of DARTS' supernet optimization. In contrast, we draw inspiration from research on ResNet (He et al., 2016) and show that this phenomenon by itself is a reasonable outcome while DARTS refines its estimation of the optimal feature map, rendering $\alpha_{skip}$ ineffective in the architecture selection.

In vanilla networks (e.g., VGG), each layer computes a new level of feature map from the output feature map of the predecessor layer; thus, reordering layers at test time would dramatically hurt the performance (Veit et al., 2016). Unlike vanilla networks, Greff et al. (2017) and Veit et al. (2016) discover that successive layers in ResNet with compatible channel sizes are in fact estimating the same optimal feature map so that the outputs of these layers stay relatively close to each other at convergence;

Table 1: Test accuracy before and after layer (edge) shuffling on cifar10. For ResNet and VGG, we randomly swap two layers in each stage (defined as successive layers between two downsampling blocks). For DARTS supernet, we randomly swap two edges in every cell.

|        | VGG             | ResNet            | DARTS            |
|--------|-----------------|-------------------|------------------|
| Before | 92.69           | 93.86             | 88.44            |
| After  | $9.83 \pm 0.33$ | $83.2015 \pm 2.03$ | $81.09 \pm 1.87$ |

As a result, ResNet's test accuracy remains robust under layer reordering. Greff et al. (2017) refers to this unique way of feature map estimation in ResNet as the "unrolled estimation."

DARTS' supernet resembles ResNet, rather than vanilla networks like VGG, in both appearance and behavior. Appearance-wise, within a cell of DARTS' supernet, edges with skip connection are in direct correspondence with the successive residual layers in ResNet. Behavior-wise, DARTS' supernet also exhibits a high degree of robustness under edge shuffling. As shown in Table 1, randomly reordering edges on a pretrained DARTS' supernet at test time also has little effect on its performance. This evidence indicates that DARTS performs unrolled estimation like ResNet as well, i.e., edges within a cell share the same optimal feature map that they try to estimate. In the following proposition, we apply this finding and provide the optimal solution of $\alpha$ in the sense of minimizing the variance of feature map estimation.

**Proposition 1.** [1] *Without loss of generality, consider one cell from a simplified search space consists of two operations: (skip, conv). Let $m^*$ denotes the optimal feature map, which is shared across all edges according to the unrolled estimation view (Greff et al., 2017). Let $o_e(x_e)$ be the output of convolution operation, and let $x_e$ be the skip connection (i.e., the input feature map of edge $e$). Assume $m^*$, $o_e(x_e)$ and $x_e$ are normalized to the same scale. The current estimation of $m^*$ can then be written as:*

$$\overline{m}_e(x_e) = \frac{\exp(\alpha_{conv})}{\exp(\alpha_{conv}) + \exp(\alpha_{skip})} o_e(x_e) + \frac{\exp(\alpha_{skip})}{\exp(\alpha_{conv}) + \exp(\alpha_{skip})} x_e, \quad (2)$$

*where $\alpha_{conv}$ and $\alpha_{skip}$ are the architecture parameters defined in DARTS. The optimal $\alpha_{conv}^*$ and $\alpha_{skip}^*$ minimizing $var(\overline{m}_e(x_e) - m^*)$, the variance of the difference between the optimal feature map $m^*$ and its current estimation $\overline{m}_e(x_e)$, are given by:*

$$\alpha_{conv}^* \propto var(x_e - m^*) \quad (3)$$

$$\alpha_{skip}^* \propto var(o_e(x_e) - m^*). \quad (4)$$

We refer the reader to Appendix A.4 for detailed proof. From eq. (3) and eq. (4), we can see that the relative magnitudes of $\alpha_{skip}$ and $\alpha_{conv}$ come down to which one of $x_e$ or $o_e(x_e)$ is closer to $m^*$ in variance:

- $x_e$ (input of edge $e$) comes from the mixed output of the previous edge. Since the goal of every edge is to estimate $m^*$ (unrolled estimation), $x_e$ is also directly estimating $m^*$.

---

[1]Proposition 1 unfolds the optimal $\alpha$ in principle and does not constraint the particular optimization method (i.e., bilevel, single-level, or blockwise update) to achieve it. Moreover, this proposition can be readily extended to various other search spaces since we can group all non-skip operations into a single $o_e(\cdot)$.

- $o_e(x_e)$ is the output of a single convolution operation instead of the complete mixed output of edge $e$, so it will deviate from $m^*$ even at convergence.

Therefore, in a well-optimized supernet, $x_e$ will naturally be closer to $m^*$ than $o_e(x_e)$, causing $\alpha_{skip}$ to be greater than $\alpha_{conv}$.

Our analysis above indicates that the better the supernet, the larger the $(\alpha_{skip} - \alpha_{conv})$ gap (softmaxed) will become since $x_e$ gets closer and closer to $m^*$ as the supernet is optimized. This result is evidenced in Figure 3, where $mean(\alpha_{skip} - \alpha_{conv})$ continues to grow as the supernet gets better. In this case, although $\alpha_{skip} > \alpha_{conv}$ is reasonable by itself, it becomes an inductive bias to NAS if we were to select the final architecture based on $\alpha$.

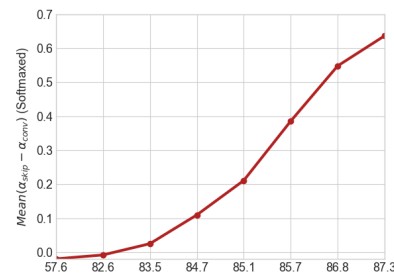

## 4 PERTURBATION-BASED ARCHITECTURE SELECTION

Figure 3: $mean(\alpha_{skip} - \alpha_{conv})$ (softmaxed) v.s. supernet's validation accuracy. The gap of $(\alpha_{skip} - \alpha_{conv})$ increases as supernet gets better.

Instead of relying on the $\alpha$ value to select the best operation, we propose to directly evaluate operation strength in terms of its contribution to the supernet's performance. The operation selection criterion is laid out in section 4.1. In section 4.2, we describe the entire architecture selection process.

### 4.1 EVALUATING THE STRENGTH OF EACH OPERATION

In section 3.1, we define the strength of each operation on a given edge as how much it contributes to the performance of the supernet, measured by discretization accuracy. To avoid inaccurate evaluation due to large disturbance of the supernet during discretization, we fine-tune the remaining supernet until it converges again, and then compute its validation accuracy (discretization accuracy at convergence). The fine-tuning process needs to be carried out for evaluating each operation on an edge, leading to substantial computation costs.

To alleviate the computational overhead, we consider a more practical measure of operation strength: for each operation on a given edge, we mask it out while keeping all other operations, and re-evaluate the supernet. The one that results in the largest drop in the supernet's validation accuracy will be considered as the most important operation on that edge. This alternative criterion incurs much less perturbation to the supernet than discretization since it only deletes one operation from the supernet at a time. As a result, the supernet's validation accuracy after deletion stays close to the unmodified supernet, and thus it alleviates the requirement of tuning the remaining supernet to convergence. Therefore, we implement this measurement for the operation selection in this work.

---

**Algorithm 1:** Perturbation-based Architecture Selection

**Input:** A pretrained supernet $S$, Set of edges $\mathcal{E}$ from $S$, Set of nodes $\mathcal{N}$ from $S$
**Result:** Set of selected operations $\{o_e^*\}_{e \in \mathcal{E}}$
**while** $|\mathcal{E}| > 0$ **do**
    randomly select an edge $e \in \mathcal{E}$ (and remove it from $\mathcal{E}$);
    **forall** *operation $o$ on edge $e$* **do**
        evaluate the validation accuracy of $S$ when $o$ is removed ($ACC_{\backslash o}$);
    **end**
    select the best operation for $e$: $o_e^* \leftarrow \arg\min_o ACC_{\backslash o}$;
    discretize edge $e$ to $o_e^*$ and tune the remaining supernet for a few epochs;
**end**

---

### 4.2 THE COMPLETE ARCHITECTURE SELECTION PROCESS

Our method operates directly on top of DARTS' pretrained supernet. Given a supernet, we randomly iterate over all of its edges. We evaluate each operation on an edge, and select the best one to be discretized based on the measurement described in section 4.1. After that, we tune the supernet for

a few epochs to recover the accuracy lost during discretization. The above steps are repeated until all edges are decided. Algorithm 1 summarizes the operation selection process. The cell topology is decided in a similar fashion. We refer the reader to Appendix A.3 for the full algorithm, including deciding the cell topology. This simple method is termed "perturbation-based architecture selection (PT)" in the following sections.

## 5 EXPERIMENTAL RESULTS

In this section, we demonstrate that the perturbation-based architecture selection method is able to consistently find better architectures than those selected based on the values of $\alpha$. The evaluation is based on the search space of DARTS and NAS-Bench-201 (Dong & Yang, 2020), and we show that the perturbation-based architecture selection method can be applied to several variants of DARTS.

### 5.1 RESULTS ON DARTS' CNN SEARCH SPACE

We keep all the search and retrain settings identical to DARTS since our method only modifies the architecture selection part. After the search phase, we perform perturbation-based architecture selection following Algorithm 1 on the pretrained supernet. We tune the supernet for 5 epochs between two selections as it is enough for the supernet to recover from the drop of accuracy after discretization. We run the search and architecture selection phase with four random seeds and report both the best and average test errors of the obtained architectures.

As shown in Table 2, the proposed method (DARTS+PT) improves DARTS' test error from 3.00% to 2.61%, with manageable search cost (0.8 GPU days). Note that by only changing the architecture selection method, DARTS performs significantly better than many other differentiable NAS methods that enjoy carefully designed optimization process of the supernet, such as GDAS (Dong & Yang, 2019) and SNAS (Xie et al., 2019). This empirical result suggests that architecture selection is crucial to DARTS: with the proper selection algorithm, DARTS remains a very competitive method.

Our method is also able to improve the performance of other variants of DARTS. To show this, we evaluate our method on SDARTS(rs) and SGAS (Chen & Hsieh, 2020; Li et al., 2020). SDARTS(rs) is a variant of DARTS that regularizes the search phase by applying Gaussian perturbation to $\alpha$. Unlike DARTS and SDARTS, SGAS performs progressive search space shrinking. Concretely, SGAS progressively discretizes its edges with the order from most to least important, based on a novel edge importance score. For a fair comparison, we keep its unique search space shrinking process unmodified and only replace its magnitude-based operation selection with ours. As we can see from Table 2, our method consistently achieves better average test errors than its magnitude-based counterpart. Concretely, the proposed method improves SDARTS' test error from 2.67% to 2.54% and SGAS' test error from 2.66% to 2.56%. Moreover, the best architecture discovered in our experiments achieves a test error of 2.44%, ranked top among other NAS methods.

### 5.2 PERFORMANCE ON NAS-BENCH-201 SEARCH SPACE

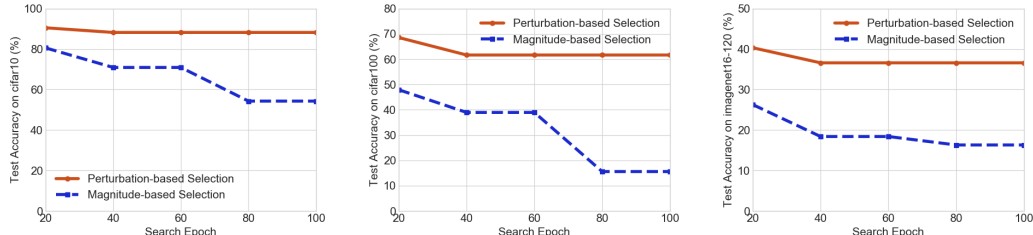

Figure 4: Trajectory of test accuracy on space NAS-Bench-201 and three datasets (Left: cifar10, Middle: cifar100, Right: Imagenet16-120). The test accuracy of our method is plotted by taking the snapshots of DARTS' supernet at corresponding epochs and run our selection method on top of it.

To further verify the effectiveness of the proposed perturbation-based architecture selection, we conduct experiments on NAS-Bench-201. NAS-Bench-201 provides a unified cell-based search space similar to DARTS. Every architecture in the search space is trained under the same protocol on three

Table 2: Comparison with state-of-the-art image classifiers on CIFAR-10.

| Architecture | Test Error (%) | Params (M) | Search Cost (GPU days) | Search Method |
|---|---|---|---|---|
| DenseNet-BC (Huang et al., 2017) | 3.46 | 25.6 | - | manual |
| NASNet-A (Zoph et al., 2018) | 2.65 | 3.3 | 2000 | RL |
| AmoebaNet-A (Real et al., 2019) | $3.34 \pm 0.06$ | 3.2 | 3150 | evolution |
| AmoebaNet-B (Real et al., 2019) | $2.55 \pm 0.05$ | 2.8 | 3150 | evolution |
| PNAS (Liu et al., 2018)* | $3.41 \pm 0.09$ | 3.2 | 225 | SMBO |
| ENAS (Pham et al., 2018) | 2.89 | 4.6 | 0.5 | RL |
| NAONet (Luo et al., 2018) | 3.53 | 3.1 | 0.4 | NAO |
| SNAS (moderate) (Xie et al., 2019) | $2.85 \pm 0.02$ | 2.8 | 1.5 | gradient |
| GDAS (Dong & Yang, 2019) | 2.93 | 3.4 | 0.3 | gradient |
| BayesNAS (Zhou et al., 2019) | $2.81 \pm 0.04$ | 3.4 | 0.2 | gradient |
| ProxylessNAS (Cai et al., 2019)[†] | 2.08 | 5.7 | 4.0 | gradient |
| NASP (Yao et al., 2020) | $2.83 \pm 0.09$ | 3.3 | 0.1 | gradient |
| P-DARTS (Chen et al., 2019) | 2.50 | 3.4 | 0.3 | gradient |
| PC-DARTS (Xu et al., 2020) | $2.57 \pm 0.07$ | 3.6 | 0.1 | gradient |
| R-DARTS (L2) Zela et al. (2020) | $2.95 \pm 0.21$ | - | 1.6 | gradient |
| DARTS (Liu et al., 2019) | $3.00 \pm 0.14$ | 3.3 | 0.4 | gradient |
| SDARTS-RS (Chen & Hsieh, 2020) | $2.67 \pm 0.03$ | 3.4 | 0.4 | gradient |
| SGAS (Cri 1. avg) (Li et al., 2020) | $2.66 \pm 0.24$ | 3.7 | 0.25 | gradient |
| DARTS+PT (avg)* | $2.61 \pm 0.08$ | 3.0 | 0.8[‡] | gradient |
| DARTS+PT (best) | 2.48 | 3.3 | 0.8[‡] | gradient |
| SDARTS-RS+PT (avg)* | $2.54 \pm 0.10$ | 3.3 | 0.8[‡] | gradient |
| SDARTS-RS+PT (best) | 2.44 | 3.2 | 0.8[‡] | gradient |
| SGAS+PT (Crit.1 avg)* | $2.56 \pm 0.10$ | 3.9 | 0.29[‡] | gradient |
| SGAS+PT (Crit.1 best) | 2.46 | 3.9 | 0.29[‡] | gradient |

[†] Obtained on a different space with PyramidNet (Han et al., 2017) as the backbone.

[‡] Recorded on a single GTX 1080Ti GPU.

* Obtained by running the search and retrain phase under four different seeds and computing the average test error of the derived architectures.

datasets (cifar10, cifar100, and imagenet16-120), and their performance can be obtained by querying the database. As in section 5.1, we take the pretrained supernet from DARTS and apply our method on top of it. All other settings are kept unmodified. Figure 4 shows the performance trajectory of DARTS+PT compared with DARTS. While the architectures found by magnitude-based selection degenerates over time, the perturbation-based method is able to extract better architectures from the same underlying supernets stably. The result implies that the DARTS' degenerated performance comes from the failure of magnitude based architecture selection.

# 6 ANALYSIS

## 6.1 ISSUE WITH THE ROBUSTNESS OF DARTS

Zela et al. (2020) observes that DARTS tends to yield degenerate architectures with abysmal performance. We conjecture that this robustness issue of DARTS can be explained by the failure of magnitude-based architecture selection.

To show this, we test DARTS' performance with perturbation-based architecture selection on four spaces proposed by Zela et al. (2020) (S1-S4). The complete specifications of these spaces can be found in Appendix A.2. Given a supernet, the architecture selected based on $\alpha$ performs poorly across spaces and datasets (column 3 in Table 3). However, our method is able to consistently extract meaningful architectures with significantly improved performance (Column 4 in Table 3).

Table 3: DARTS+PT on S1-S4 (test error (%)).

| Dataset | Space | DARTS | DARTS+PT (Ours) | DARTS+PT (fix $\alpha$)* |
|---|---|---|---|---|
| C10 | S1 | 3.84 | 3.50 | 2.86 |
| | S2 | 4.85 | 2.79 | 2.59 |
| | S3 | 3.34 | 2.49 | 2.52 |
| | S4 | 7.20 | 2.64 | 2.58 |
| C100 | S1 | 29.46 | 24.48 | 24.40 |
| | S2 | 26.05 | 23.16 | 23.30 |
| | S3 | 28.90 | 22.03 | 21.94 |
| | S4 | 22.85 | 20.80 | 20.66 |
| SVHN | S1 | 4.58 | 2.62 | 2.39 |
| | S2 | 3.53 | 2.53 | 2.32 |
| | S3 | 3.41 | 2.42 | 2.32 |
| | S4 | 3.05 | 2.42 | 2.39 |

* This column will be explained later in Section 6.3

Notably, DARTS+PT is able to find meaningful architecture on S2 (*skip_connect, sep_conv_3x3*) and S4 (*noise, sep_conv_3x3*), where DARTS failed dramatically. As shown in Figure 5, on S2, while magnitude-based selection degenerates to architectures filled with skip connections, DARTS+PT is

able to find architecture with 4 convolutions; On S4, DARTS+PT consistently favors *sep_conv_3x3* on edges where $\alpha$ selects *noise*.

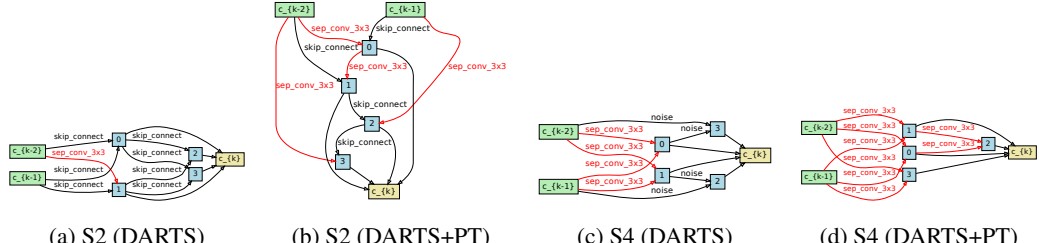

| (a) S2 (DARTS) | (b) S2 (DARTS+PT) | (c) S4 (DARTS) | (d) S4 (DARTS+PT) |

Figure 5: Comparison of normal cells found on S2 and S4. The perturbation-based architecture selection (DARTS+PT) is able to find reasonable architectures in cases where the magnitude-based method (DARTS) fails dramatically. The complete architecture can be found in Appendix A.9. Non-trivial operations are marked red.

## 6.2 PROGRESSIVE TUNING

In addition to operation selection, we also tune the supernet after an edge is discretized so that the supernet could regain the lost accuracy. To measure the effectiveness of our operation selection criterion alone, we conduct an ablation study on the progressive tuning part. Concretely, we test a baseline by combining progressive tuning with magnitude-based operation selection instead of our selection criterion, which we code-named DARTS+PT-Mag. Figure 6 plots the change of validation accuracy of DARTS+PT and DARTS+PT-Mag during the operation selection phase. As we can see, DARTS+PT is able to identify better operations that lead to higher validation accuracy than the magnitude-based alternative, revealing the effectiveness of our operation selection criteria. Moreover, DARTS+PT-Mag is only able to obtain a test error of 2.85% on DARTS space

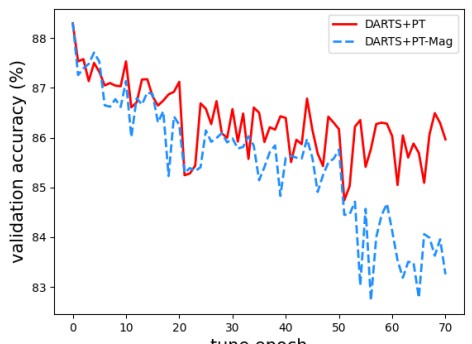

Figure 6: The trajectory of validation accuracy in the operation selection phase on S2. DARTS+PT is able to select better operations that lead to higher accuracy of the supernet than DARTS+PT-Mag.

on cifar10, much worse than DARTS+PT (2.61%), indicating that the operation selection part plays a crucial role in our method.

## 6.3 FIXING $\alpha$ AS UNIFORM

Since the proposed method does not rely on $\alpha$ for architecture selection, a natural question is whether it is necessary to optimize a stand-alone $\alpha$. We find that by fixing $\alpha = 0$ (uniform weights for all the operations) while training supernet and applying perturbation-based architecture se-

Table 4: DARTS+PT v.s. DARTS+PT (fixed $\alpha$) on more spaces (test error %) on cifar10.

| Space | DARTS | DARTS+PT | DARTS+PT (fix $\alpha$) |
|---|---|---|---|
| DARTS Space | 3.00 | 2.61 | 2.87 |
| NAS-Bench-201 | 45.7 | 11.89 | 6.20 |

lection, the resulting method performs on-par with DARTS+PT, and in some cases even better. For example, DARTS+PT (fix $\alpha$) achieves better performance than DARTS+PT on NAS-Bench-201. On DARTS' search space and its variants S1-S4, DARTS+PT (fix $\alpha$) performs similarly to DARTS+PT. The results can be found in Table 3 and Table 4. This surprising finding suggests that even the most naive approach, simply training a supernet without $\alpha$, will be a competitive method when combining with the proposed perturbation-based architecture selection.

## 7 CONCLUSION AND DISCUSSION

This paper attempts to understand Differentiable NAS methods from the architecture selection perspective. We re-examine the magnitude-based architecture selection process of DARTS and provide empirical and theoretical evidence on why it does not indicate the underlying operation strength. We introduce an alternative perturbation-based architecture selection method that directly measures the operation strength via its contribution to the supernet performance. The proposed selection method is able to consistently extract improved architecture from supernets trained identically to the respective base methods on several spaces and datasets.

Our method brings more freedom in supernet training as it does not rely on $\alpha$ to derive the final architecture. We hope the perturbation-based architecture selection can bring a new perspective to the NAS community to rethink the role of $\alpha$ in Differential NAS.

## ACKNOWLEDGEMENT

This work is supported in part by NSF under IIS-1901527, IIS-2008173, IIS-2048280 and by Army Research Laboratory under agreement number W911NF-20-2-0158.

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

# A  APPENDIX

## A.1  DESCRIPTION ABOUT OUR BASELINE MODELS

### A.1.1  DARTS

DARTS (Liu et al., 2019) is a pioneering work that introduces the general differentiable NAS framework, which we reviewed in section 2. In DARTS, the topology and operation are searched together. Concretely, at the end of the search, it selects one operation for every edge in the normal (reduction) cell based on the architecture parameter $\alpha$. Then it selects two input edges for each node in the cell by comparing the largest $\alpha$ of every input edge. The final architecture consists of one operation on each of the eight edges in the normal (reduction) cell. The operation on an edge will be selected from a pool of seven candidates: skip_connection, avg_pool_3x3, max_pool_3x3, sep_conv_3x3, sep_conv_5x5, dil_conv_3x3, and dil_conv_5x5. In addition to these operations, DARTS also maintains a "none" op, which is used exclusively for determining the topology rather than treated as an operation (Liu et al., 2019). Since the main focus of our paper is operation assignment, we omit none op when applying the proposed selection method on DARTS.

### A.1.2  SDARTS

SDARTS (Chen & Hsieh, 2020) is a variant of DARTS aiming at regularizing the bilevel optimization process in DARTS via random Gaussian perturbation, inspired by the recent finding that regularizing DARTS' supernet leads to improved performance. While the optimization of architecture parameter $\alpha$ in SDARTS is identical to DARTS, it distorts the architecture parameters $\alpha$ with random Gaussian noise while training the model weights $w$. This simple yet effective regularizer is able to consistently improve the robustness and performance of DARTS.

### A.1.3  SGAS

SGAS (Li et al., 2020) represents a line of research on improving the search efficiency of differentiable NAS by progressively shrinking the search space. It first trains the model weights $w$ alone for 10 epochs. After that, it selects one edge from the supernet and then selects the best operation on that edge based on $\alpha$ to be discretized. The edge selection is based on the ranking of the proposed edge importance score. The process stops after all eight edges of the final architecture are decided.

## A.2  MICROARCHITECTURE OF SPACE S1 - S4

Zela et al. (2020) introduces four variants of the DARTS' space (S1, S2, S3, and S4) to study the robustness of DARTS. These spaces differ from DARTS' original space only in the number and types of operations on each edge. Apart from that, everything else is the same.

- S1 is a pre-optimized search space consisting of top2 operations selected by DARTS. As a result, each edge contains a different set of operations to be searched from.

- S2 consists of two operations: *skip_connect* and *sep_conv_3x3*.

- S3 consists of three operations: *none*, *skip_connect* and *sep_conv_3x3*.

- S4 consists of two operations: *noise* and *sep_conv_3x3*. The *noise* operation outputs a random Gaussian noise $\mathcal{N}(0, 1)$ regardless of the input. This operation generally hurts the performance of discretized architecture, and should be avoided by NAS algorithm.

## A.3 THE COMPLETE ALGORITHM OF PERTURBATION-BASED ARCHITECTURE SELECTION

---

**Algorithm 2:** Perturbation-based Architecture Selection

---

**Input:** A pretrained Supernet $S$, Set of Edges $\mathcal{E}$ from $\mathcal{S}$, Set of Nodes $\mathcal{N}$ from $\mathcal{S}$
**Result:** Set of selected operations $\{o_e^*\}_{e\in\mathcal{E}}$, and top2 input edges for each node
  $\{(e_n^{(1)*}, e_n^{(2)*})\}_{n\in\mathcal{N}}$
**while** $|\mathcal{E}| > 0$ **do** // operation selection phase
  randomly select an edge $e \in \mathcal{E}$ (and remove it from $\mathcal{E}$);
  **forall** *operation $o$ on edge $e$* **do**
    | evaluate the validation accuracy of $S$ when $o$ is removed ($ACC_{\backslash o}$);
  **end**
  select the best operation on $e$: $o_e^* \leftarrow \arg\min_o ACC_{\backslash o}$;
  discretize edge $e$ to $o_e^*$ and train the remaining supernet until it converges again;
**end**
**while** $|\mathcal{N}| > 0$ **do** // topology selection phase
  randomly select a node $n \in \mathcal{N}$ (and remove it from $\mathcal{N}$);
  **forall** *input edge $e$ of node $n$* **do**
    | evaluate the validation accuracy of $S$ when $e$ is removed ($ACC_{\backslash e}$);
  **end**
  set top2 edges on $n$ $(e_n^{(1)*}, e_n^{(2)*})$ to be the ones with lowest and second lowest $ACC_{\backslash e}$;
  prune out all other edges of $n$ and train the remaining supernet until it converges again;
**end**

---

## A.4 PROOF OF PROPOSITION 1

*Proof.* Let $\theta_{skip} = Softmax(\alpha_{skip})$ and $\theta_{conv} = Softmax(\alpha_{conv})$. Then the mixed operation can be written as $\overline{m}_e(x_e) = \theta_{conv}o_e(x_e) + \theta_{skip}x_e$. We formally formulate the objective to be:

$$\min_{\theta_{skip},\theta_{conv}} Var(\overline{m}_e(x_e) - m^*) \tag{5}$$

$$s.t. \quad \theta_{skip} + \theta_{conv} = 1 \tag{6}$$

This constraint optimization problem can be solved with Lagrangian multipliers:

$$L(\theta_{skip}, \theta_{conv}, \lambda) = Var(\overline{m}_e(x_e) - m^*) - \lambda(\theta_{skip} + \theta_{conv} - 1) \tag{7}$$

$$= Var(\theta_{conv}o_e(x_e) + \theta_{skip}x_e - m^*) - \lambda(\theta_{skip} + \theta_{conv} - 1) \tag{8}$$

$$= Var(\theta_{conv}(o_e(x_e) - m^*) + \theta_{skip}(x_e - m^*))$$
$$- \lambda(\theta_{skip} + \theta_{conv} - 1) \tag{9}$$

$$= Var(\theta_{conv}(o_e(x_e) - m^*)) + Var(\theta_{skip}(x_e - m^*))$$
$$+ 2Cov(\theta_{conv}(o_e(x_e) - m^*), \theta_{skip}(x_e - m^*))$$
$$- \lambda(\theta_{skip} + \theta_{conv} - 1) \tag{10}$$

$$= \theta_{conv}^2 Var(o_e(x_e) - m^*) + \theta_{skip}^2 Var(x_e - m^*)$$
$$+ 2\theta_{conv}\theta_{skip}Cov(o_e(x_e) - m^*, x_e - m^*)$$
$$- \lambda(\theta_{skip} + \theta_{conv} - 1) \tag{11}$$

Setting:

$$\frac{\partial L}{\partial \lambda} = \theta_{conv} + \theta_{skip} - 1 = 0 \tag{12}$$

$$\frac{\partial L}{\partial \theta_{conv}} = 2\theta_{conv}Var(o_e(x_e) - m^*) + 2\theta_{skip}Cov(o_e(x_e) - m^*, x_e - m^*)$$
$$- \lambda = 0 \tag{13}$$

$$\frac{\partial L}{\partial \theta_{skip}} = 2\theta_{conv}Cov(o_e(x_e) - m^*, x_e - m^*) + 2\theta_{skip}Var(x_e - m^*)$$
$$- \lambda = 0 \tag{14}$$

Solving the above equations will give us:

$$(15)$$

$$\theta^*_{conv} = \frac{Var(x_e - m^*) - Cov(o_e(x_e) - m^*, x_e - m^*)}{Z} \tag{16}$$

$$\theta^*_{skip} = \frac{Var(o_e(x_e) - m^*) - Cov(o_e(x_e) - m^*, x_e - m^*)}{Z} \tag{17}$$

Where $Z = Var(o_e(x_e) - m^*) - Cov(o_e(x_e) - m^*, x_e - m^*) + Var(x_e - m^*) - Cov(o_e(x_e) - m^*, x_e - m^*)$. Aligning basis with DARTS, we get:

$$\alpha^*_{conv} = \log\left[Var(x_e - m^*) - Cov(o_e(x_e) - m^*, x_e - m^*)\right] + C \tag{18}$$

$$\alpha^*_{skip} = \log\left[Var(o_e(x_e) - m^*) - Cov(o_e(x_e) - m^*, x_e - m^*)\right] + C \tag{19}$$

The only term that differentiates $\alpha_{skip}$ from $\alpha_{conv}$ is the first term inside the logarithm, therefore:

$$\alpha^*_{conv} \propto var(x_e - m^*) \tag{20}$$

$$\alpha^*_{skip} \propto var(o_e(x_e) - m^*) \tag{21}$$

$\square$

## A.5 MORE FIGURES ON $\alpha$ AND DISCRETIZATION ACCURACY AT CONVERGENCE

We provide extra figures similar to Figure 1 to take into account the randomness of supernet's training. We first train 6 supernets with different seeds, and then randomly select 1 edge from each of them. We can see that the results are consistent with Figure 1. As shown in Figure 7, the magnitude of $\alpha$ for each operation does not necessarily agree with its relative discretization accuracy at convergence.

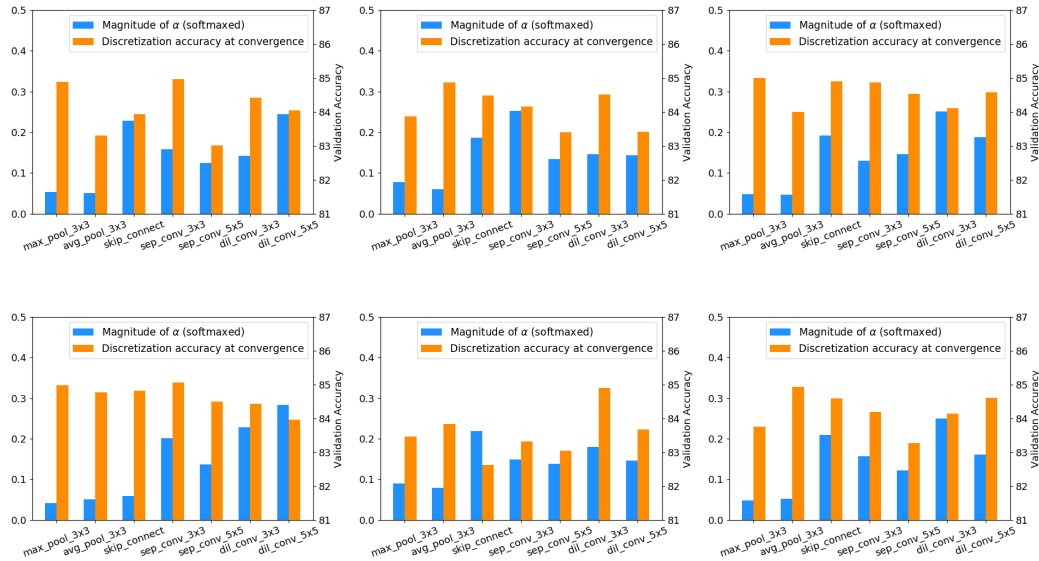

Figure 7: $\alpha$ v.s. discretization accuracy at convergence of all the operations on 6 randomly selected edges from DARTS' supernet trained with different seeds (one subfigure for each edge).

## A.6 PERFORMANCE TRAJECTORY OF DARTS+PT (FIX $\alpha$) ON NAS-BENCH-201

We plot the performance trajectory of DARTS+PT (fix $\alpha$) on NAS-Bench-201 similar to Figure 4. As shown in Figure 8, it consistently achieves strong performance without training $\alpha$ at all, indicating that the extra freedom of supernet training without $\alpha$ can be explored to develop improved search algorithm in the future.

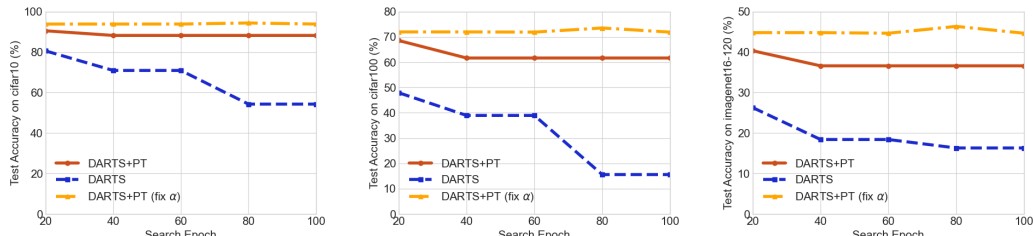

Figure 8: Trajectory of test accuracy of architectures found by DARTS+PT (fix $\alpha$) on space NAS-Bench-201 and 3 datasets (Left: cifar10, Middle: cifar100, Right: Imagenet16-120).

Table 5: Evaluation of the derived architecture on ImageNet in the mobile setting.

| Architecture | Test Error(%) | | Params | Search |
|---|---|---|---|---|
| | top-1 | top-5 | (M) | Method |
| DARTS (Liu et al., 2019) | 26.7 | 8.7 | 4.7 | gradient |
| DARTS+PT | 25.5 | 8.0 | 4.6 | gradient |

## A.7 ABLATION STUDY ON THE NUMBER OF FINE-TUNING EPOCHS

As described in section 4.2, we perform fine-tuning between two edge decisions to recover supernet from the accuracy drop after discretization. The number of fine-tuning epochs is set to 5 for all experiments because empirically we find that it is enough for the supernet to converge again. In this section, we conduct an ablation study on the effect of the number of fine-tuning epochs. As shown in Figure 9, the gain from tuning the supernet longer than 5 epochs is marginal.

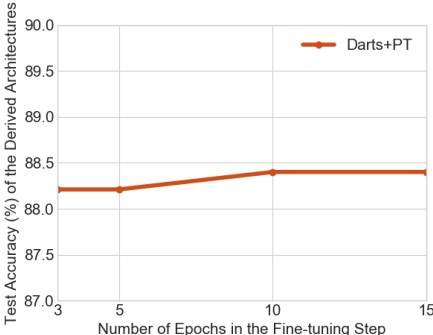

Figure 9: Performance of DARTS+PT under different number of fine-tuning epochs on NAS-Bench-201. Tuning the supernet longer results in marginal improvement on the proposed method.

## A.8 ARCHITECTURE TRANSFERABILITY EVALUATION ON IMAGENET

We further evaluate the performance of the derived architecture on ImageNet. We strictly follow the training protocals as well as the hyperparameter settings of DARTS (Liu et al., 2019) for this experiment. As shown in Table 5, the proposed method improves the top1 performance of DARTS by 1.2%.

## A.9 SEARCHED ARCHITECTURES

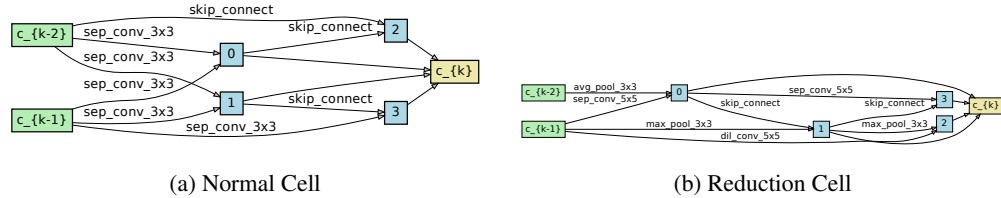

(a) Normal Cell                                    (b) Reduction Cell

Figure 10: Normal and Reduction cells discovered by DARTS+PT on cifar10

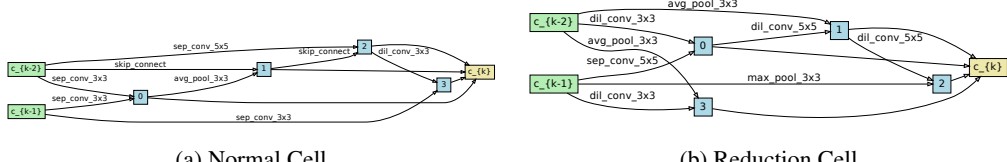

(a) Normal Cell                                    (b) Reduction Cell

Figure 11: Normal and Reduction cells discovered by SDARTS+PT on cifar10

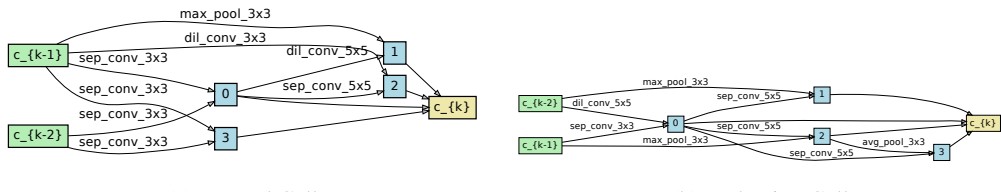

(a) Normal Cell                                    (b) Reduction Cell

Figure 12: Normal and Reduction cells discovered by SGAS-PT on cifar10

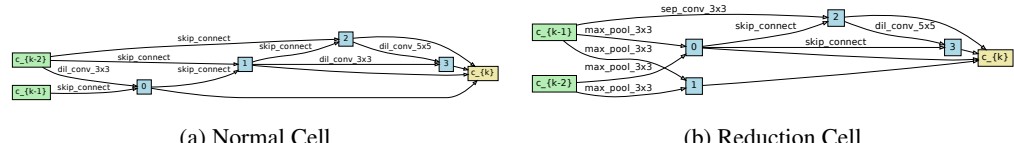

(a) Normal Cell                                    (b) Reduction Cell

Figure 13: Normal and Reduction cells discovered by DARTS+PT on cifar10 on Space S1

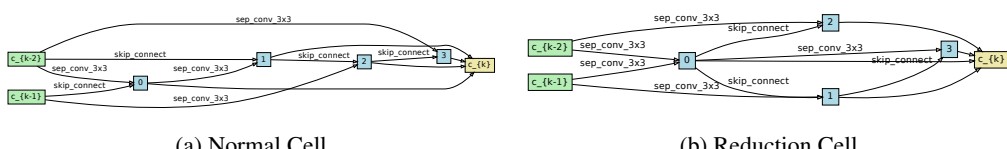

(a) Normal Cell                                    (b) Reduction Cell

Figure 14: Normal and Reduction cells discovered by DARTS+PT on cifar10 on Space S2

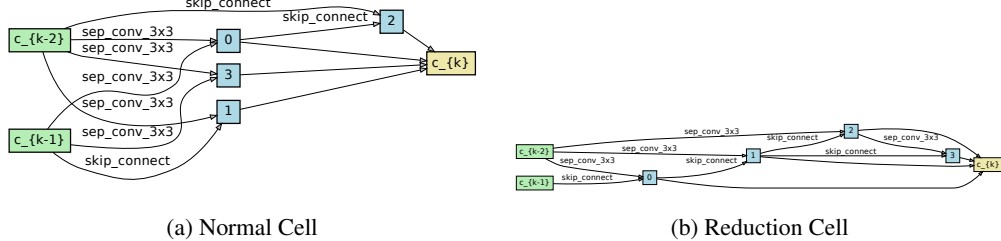

(a) Normal Cell                                    (b) Reduction Cell

Figure 15: Normal and Reduction cells discovered by DARTS+PT on cifar10 on Space S3

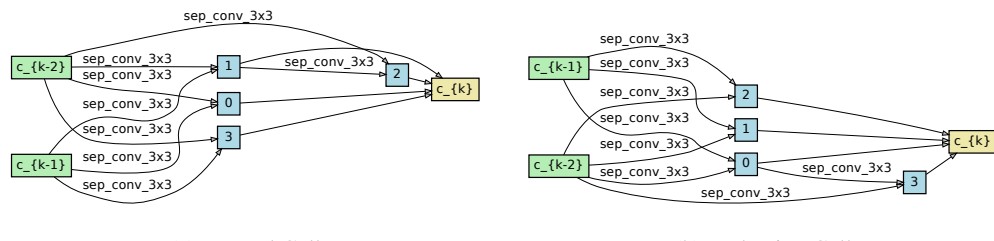

(a) Normal Cell                                    (b) Reduction Cell

Figure 16: Normal and Reduction cells discovered by DARTS+PT on cifar10 on Space S4

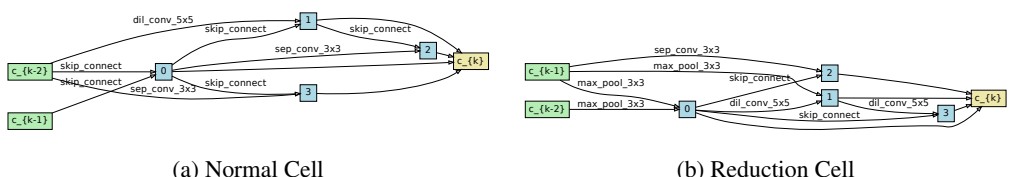

(a) Normal Cell                                    (b) Reduction Cell

Figure 17: Normal and Reduction cells discovered by DARTS+PT on cifar100 on Space S1

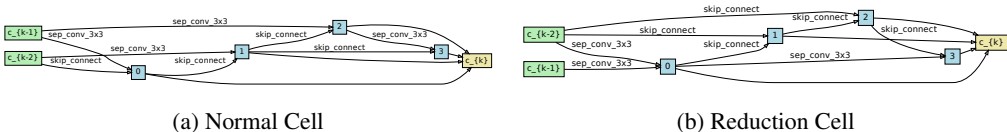

(a) Normal Cell                                    (b) Reduction Cell

Figure 18: Normal and Reduction cells discovered by DARTS+PT on cifar100 on Space S2

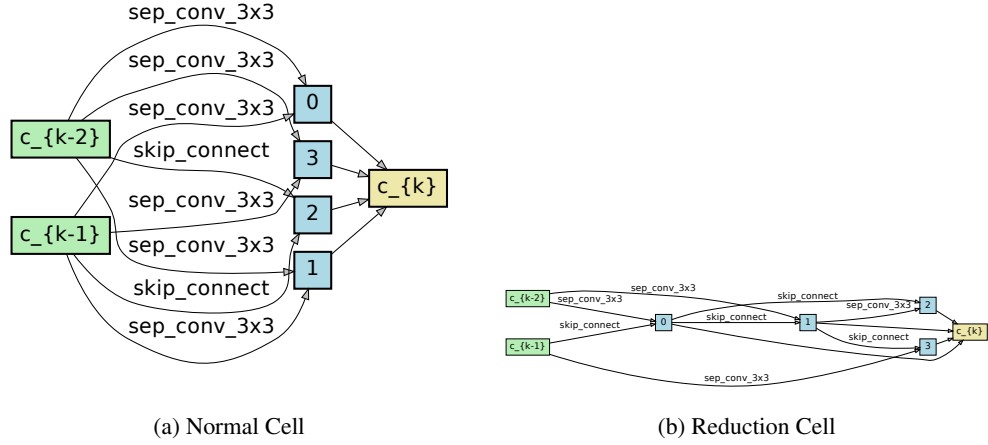

(a) Normal Cell                                    (b) Reduction Cell

Figure 19: Normal and Reduction cells discovered by DARTS+PT on cifar100 on Space S3

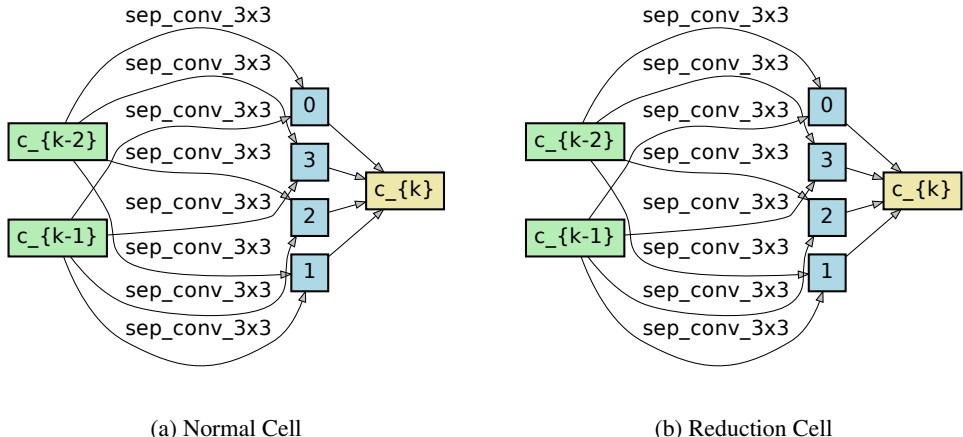

(a) Normal Cell             (b) Reduction Cell

Figure 20: Normal and Reduction cells discovered by DARTS+PT on cifar100 on Space S4

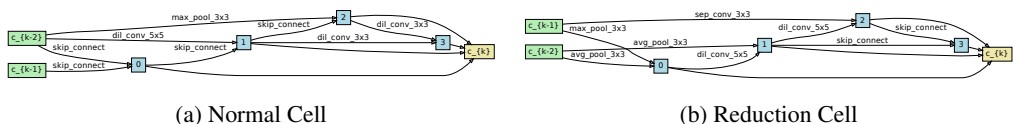

(a) Normal Cell             (b) Reduction Cell

Figure 21: Normal and Reduction cells discovered by DARTS+PT on svhn on Space S1

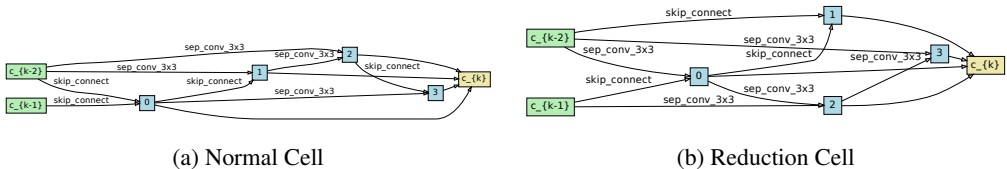

(a) Normal Cell             (b) Reduction Cell

Figure 22: Normal and Reduction cells discovered by DARTS+PT on svhn on Space S2

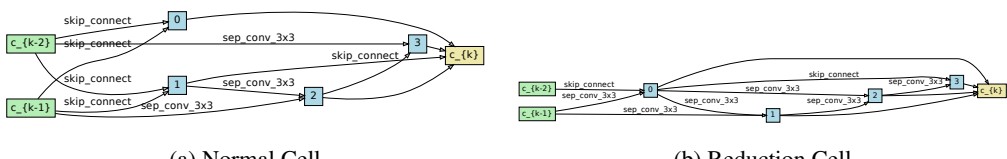

(a) Normal Cell             (b) Reduction Cell

Figure 23: Normal and Reduction cells discovered by DARTS+PT on svhn on Space S3

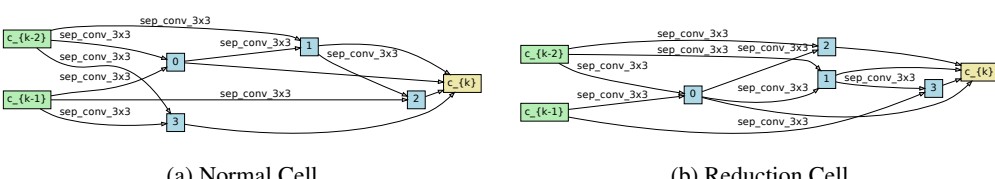

(a) Normal Cell             (b) Reduction Cell

Figure 24: Normal and Reduction cells discovered by DARTS+PT on svhn on Space S4

