# OpenReview forum: "Rethinking Architecture Selection in Differentiable NAS"
_ICLR.cc/2021/Conference — ICLR 2021 Oral_

### Official Review · AnonReviewer3 · 2020-10-24
**The work analyzes the differential NAS methods from a new perspective that the value alpha is not suitable for selecting edges. Based on this observation, the author proposes a new method to evaluate the edge strength.**

**Rating:** 7
**Confidence:** 4

**Review:**

Pros
- The work analyzes the differential NAS methods from a new perspective that the value alpha is not suitable for selecting edges. Based on this observation, the author proposes a new method to evaluate the edge strength.
- The motivation is clear and the finding of the residual path is interesting.
- The proposed method is effective and shows good results.

Cons
- All experiments are based on the Cifar10 dataset, the author may consider extending the proposed method to a larger dataset such as ImageNet.
- The efficiency of the proposed method, what is the computational cost?
- Some descriptions and figures are not very clear.  E.g., in Figure 1, what do the three figures stand for respectively?
- Except for the residual connection, can the same rules be found in other layers?

The paper is interesting and has found some values for the NAS community by rethinking the representation ability of the important factor $\alpha$. The paper may need proof-reading and do some experiments on a more widely used large-scale dataset. I tend to accept this paper.

---

> ### Author Response · Authors · 2020-11-19
> **Response to AnonReviewer3**
>
> Thank you for your positive comments. We respond to your questions and concerns below.
>
> *1. "All experiments are based on the Cifar10 dataset, the author may consider extending the proposed method to a larger dataset such as ImageNet."*
>
> Thank you for your suggestion. We include an experiment on the ImageNet test accuracy of the derived architecture. It can be found in Table 5 in Appendix A.8 of the revised manuscript. Our proposed method improves the test error on ImageNet of Darts from 26.7% to 25.5%.
> \
> \
> *2. "The efficiency of the proposed method, what is the computational cost?"*
>
> The search cost of the proposed method can be found in Table 2. For Darts+PT, the total cost is 0.8 GPU days, which includes 0.4 GPU days to train the supernet like Darts, and 0.4 GPU days to perform the proposed selection method. So overall the algorithm is efficient.
> \
> \
> *3. “Some descriptions and figures are not very clear. E.g., in Figure 1, what do the three figures stand for respectively?”*
>
> Thank you for pointing it out. We went through several rounds of proof-reading and improved the writing. We will continue to refine the paper. The three subfigures in Figure 1 show the operation strength on three randomly selected edges (i.e., one subfigure for one random edge). Take one subfigure as an example. As marked in the legend, the blue bars represent $Softmax(\alpha)$ of 7 operations on an edge, and the orange bars represent the discretization accuracy at convergence of each operation (defined in section 3.1). Note that we add extra figures similar to Figure 1 on more randomly selected edges and seeds in Figure 7 in Appendix A.5.
> \
> \
> *4. "Except for the residual connection, can the same rules be found in other layers?"*
>
> The analysis in section 3.2 targets residual operation. It is because 1). every edge on the most popular spaces like Darts and NAS-Bench-201’s search space contains a skip connection operation, and 2). the skip connection domination issue is found to be particularly troublesome in differentiable NAS [1].
>
> On the other hand, the general analysis of the failure of magnitude-based architecture selection and the effectiveness of the proposed method is not limited to skip connections. We demonstrate that the proposed method can extract better architectures consistently on various spaces (NAS-Bench-201, Darts, and S1-S4), base methods (Darts, SDarts, and SGAS), and datasets (e.g., cifar10, cifar100, and svhn).
> \
> \
> **References**
> 1. Zela et al. Understanding and Robustifying Differentiable Architecture Search. ICLR2020

---

### Official Review · AnonReviewer2 · 2020-10-27
**Very interesting analysis on DARTS skip-connection problem, but with some issues on the proposed method**

**Rating:** 7
**Confidence:** 4

**Review:**

# post rebuttal

I have no further concerns and increase the rate to accept.

# Summary

This paper identifies an interesting phenomenon that on DARTS based method, the operation can not be simply chosen based on the maximum value of trained weights. The authors propose a new selection paradigm. For each operation, one should first discretize the soft weights encoding into a one-hot, then fine-tune the network for some iterations, and use the performance as a metric to select the operation.
It provides an interesting and novel analysis on the question of why DARTS always converges to skip-connections. The experiments are conducted on CIFAR-10, 100, and SVHN on DARTS space and NASBench-201 over three datasets.


# Strength

I enjoyed reading this paper although there are some flaws in terms of presentation. It tackles one of the most critical problems in the DARTS domain, why the skip-connection dominates after the super-net training converges. All earlier works propose some solutions in an ad-hoc manner, by early stopping. This is the first time I have seen a simple and reasonable explanation of this phenomenon, and by itself is a great contribution to the NAS community. The toy example constructed in section 3 Figure 3 also evidences this explanation, though the theoretical part only discussed a simple case with two operations.


# Weakness
I am looking forwards to hearing back from the authors and improve my scores if these issues are fixed/explained.

## Effectiveness of the method
The proposed solution is simple yet effective, though there is not another analysis of why discretization and fine-tune will triumph except comparison with baselines. On page 6 we can see the perturbation based method, test accuracy drops for both your method and the baseline after 20 epochs. Does it mean the current approach is still not working well? Training for longer simply destroys the search and without proper training, the network is essentially in a random search state, i.e. it goes back to the game that DARTS do not out-perform random search. This trend is consistent over three datasets.

## Other questions

- Ablation on the finetuning. In the algorithm part, you mentioned to train for a few epochs, but do you have some rule for that? I saw you doing this on DARTS space (Figure 6), but will this be more reasonable on NASBench-201?

- Table 1:  Will it because the swapped edges belong to the same node? For example, you could control the swap that it only happens, e.g. edge 0->2 and 2->4?


- Discretization accuracy is not clearly defined. Say for the operation weights in DARTS, [0.2, 0.4, 0.2, 0.2], does this mean you will directly evaluate the accuracy over [0, 1, 0, 0] given the supernet? Or first, make it one-hot and train to convergence while using the original super-net as a warmup? (I think I found it on page 3 after reading it again but please also help me to confirm.)

- Figure 1: NAS methods are not stable over one set of training, could authors provide another visualization, say training the darts super-net for multiple times, and shows the average of these training the alpha and its discretization accuracy? Showing three selected edges is not convincing.

## suggestions on the presentation
- Section 2.1, 2.2 and 2.3 can be a `\paragraph` instead of subsections. They are too short.
- Grammar issues
	- Abstract: one of the most ... methods (s)

---

> ### Author Response · Authors · 2020-11-19
> **Response to AnonReviewer2 (Part 3: Extending the theory to multiple operations)**
>
> *1 "The toy example constructed in section 3 Figure 3 also evidences this explanation, though the theoretical part only discussed a simple case with two operations."*
>
> There is a simple way to extend Proposition 1 to multiple operations, as briefly mentioned in footnote 1 (second sentence): We can group the operations other than skip connection into H(x). In this case, the advantage of $\alpha_{skip}$ over $\alpha_{group}$ still holds as the skip connection is the only identity mapping operation on the edge.
> \
> \
>  **References**
> 1. Zela et al. Understanding and Robustifying Differentiable Architecture Search. ICLR2020

---

> ### Author Response · Authors · 2020-11-19
> **Response to AnonReviewer2 (Part 2: Questions and Suggestions)**
>
>
> *1. "Ablation on the finetuning. In the algorithm part, you mentioned to train for a few epochs, but do you have some rule for that? I saw you doing this on DARTS space (Figure 6), but will this be more reasonable on NAS-Bench-201?"*
>
> The purpose of the find-tuning step is to keep the supernet converged before making the operation decision on the next edge. We find that 5 epochs are enough for the supernet to re-converge (for all search spaces used in this paper), so we simply set the fine-tuning epoch as 5 for all our experiments including NAS-Bench-201, Darts’ space, and S1-S4. We add an ablation study on the number of epochs in the fine-tuning step in Appendix A.7 on NAS-Bench-201. As shown in Figure 9, the proposed method works well with 5 epochs of fine-tuning; and tuning longer than 5 epochs improves the performance marginally.
>
> As discussed in Section 6.2, Figure 6 is part of a study on whether the perturbation-based operation selection criterion can identify better operations for the supernet than $\alpha$. It shows the trajectory of the supernet’s validation accuracy while running our algorithm, and the x-axis denotes the aggregated fine-tuning epochs. We still set the fine-tuning epoch as 5 (per edge) and there are 14 edges within a cell, which is why the total epoch is 70=14*5 (x-axis in Figure 6).
> \
> \
> *2. "Table 1: Will it because the swapped edges belong to the same node? For example, you could control the swap that it only happens, e.g. edge 0->2 and 2->4?"*
>
> Table 1 is produced by performing random edge swapping over 20 independent runs, thus any type of swappings could occur at test time. So the “robustness to edge shuffling” observed in Darts’ supernet holds under various types of edge swapping, including sequential edges (e.g., edge 0->2 and 2->4) and parallel edges (e.g., edge 1->4 and 3->4).
> \
> \
> *3. "Discretization accuracy is not clearly defined. Say for the operation weights in DARTS, [0.2, 0.4, 0.2, 0.2], does this mean you will directly evaluate the accuracy over [0, 1, 0, 0] given the supernet? Or first, make it one-hot and train to convergence while using the original super-net as a warmup? (I think I found it on page 3 after reading it again but please also help me to confirm.)"*
>
> Yes, the latter is correct. The discretization accuracy at convergence (we used to plot Figure 1) is computed by first making the given edge one-hot, and then fine-tuning the resulting supernet until it converges.
> \
> \
> *4. "Figure 1: NAS methods are not stable over one set of training, could authors provide another visualization, say training the darts super-net for multiple times, and shows the average of these training the alpha and its discretization accuracy? Showing three selected edges is not convincing."*
>
> Thank you for your suggestion. We add extra figures like Figure 1 based on randomly selected edges (from supernets trained with multiple different seeds). Note that the edges in Figure 1 are also randomly selected as described. You can find it in Figure 7 in Appendix A.5. The results agree with Figure 1. Moreover, in Figure 2, we show the comparison on every edge of the supernet on space S2. S2 only consists of two operations (skip: skip_connect and conv: sep_conv_3x3) and can cause Darts to fail dramatically due to excessive skip connections [1]. Figure 2 shows that although $\alpha_{skip}$ dominates $\alpha_{conv}$, the Darts’ supernet benefits more from discretizing to sep_conv_3x3 on half of the edges.
>
> We think that averaging over multiple seeds might not be informative in this case, because $\alpha$ on multiple runs might be quite different, and averaging them will cancel out meaningful signals from each run. So instead, we train the supernet with different seeds and randomly select the edges, which can take the randomness of Darts into consideration.
> \
> \
> *5. "Suggestions on the presentation"*
>
> Thank you for pointing them out, we proofread and improve the writing as you suggested. We will continue to refine the paper.

---

> ### Author Response · Authors · 2020-11-19
> **Response to AnonReviewer2 (Part 1: Effectiveness of the method)**
>
> Thank you for your positive comments. We respond to your concerns and questions below.
>
> *1. "The proposed solution is simple yet effective, though there is not another analysis of why discretization and fine-tune will triumph except comparison with baselines."*
>
> We think that the fine-tuning step comes naturally with the perturbation-based architecture selection. By construction, perturbation-based architecture selection uses the validation accuracy of the supernet to evaluate operations. After we make an operation decision for a certain edge, the validation accuracy of the supernet drops due to discretization, and therefore the supernet is no longer accurate in evaluating the operations for the remaining edges. In the worst case, if we discretize all edges of a supernet from Darts, the remaining network is entirely broken (~10% validation accuracy). So we always keep the supernet converged via fine-tuning before deciding the next edge. Moreover, the ablation study on the fine-tuning part in Section 6.2 shines some light on the contribution of perturbation-based selection and fine-tune separately; our analysis shows that the perturbation-based selection criterion contributes more to the performance of our algorithm.
> \
> \
> *2. "On page 6 we can see the perturbation-based method, test accuracy drops for both your method and the baseline after 20 epochs. Does it mean the current approach is still not working well? Training for longer simply destroys the search and without proper training, the network is essentially in a random search state, i.e. it goes back to the game that DARTS do not out-perform random search. This trend is consistent over three datasets."*
>
> Since the supernet defined on NAS-Bench-201 (Figure 4) is relatively small, it has already achieved a non-trivial validation accuracy at epoch 20 (77.4% on cifar10) compared with the random state (10% on cifar10). The main message we would like to convey in Figure 4 is that the proposed architecture selection approach is able to consistently extract better architectures while Darts degenerates drastically over time due to the failure of architecture selection, even though they are using exactly the same underlying supernet. So we think the minor drop of accuracy at epoch 40 does not alter this conclusion.
>
> Furthermore, when producing NAS-Bench-201, the authors used the same hyperparameters of Darts, even though the supernets and training protocols under these two spaces differ a lot. So the supernet training can be suboptimal on NAS-Bench-201 space. As a piece of evidence, we are able to improve the validation accuracy of the supernet on NAS-Bench-201 by simply fixing $\alpha$ as uniform and only training the model weights (codenamed as Darts+PT (fix $\alpha$) in our paper). Applying our proposed selection method on this supernet produces a stable and leveled trend. Please see Figure 8 of the revised paper. Moreover, the improvement brought by the proposed method is consistent on multiple search spaces (NAS-Bench-201, Darts’ space, and S1-S4), base methods (Darts, SDarts, and SGAS), and datasets (e.g., cifar10, cifar100, and svhn).

---

### Official Review · AnonReviewer4 · 2020-10-28
**Finalization step is crucial!**

**Rating:** 10
**Confidence:** 5

**Review:**

Summary:

In one-shot differentiable NAS, a supergraph is usually trained (via bilevel optimization as in DARTS, or other approximations to bilevel such as gumbel softmax, etc). After supergraph training, a final architecture is obtained by taking the operator at each edge which has the highest architecture weight magnitude. This step is usually termed as the 'finalization' step. (In DARTS the finalization step actually orders incoming edges by the max of the architecture weight magnitudes at each edge and selects the top two edges and the corresponding maximum architecture weight in them as the final operators.). This paper examines this quite ad hoc step very closely. It finds that the magnitude of architecture weights (alphas commonly in this niche literature) are misleading. It shows by careful ablation experiments that alpha magnitudes are very much not useful in selecting good operators.

By taking inspiration from the "unrolled estimation" viewpoint of ResNet prior work it shows that DARTS converging to degenerate architectures where alphas over parameters operators like skipconnect is actually to be expected when finalization step relies on the magnitude of alpha.

The paper proposes a much more intuitive finalization step which just picks the operator at each edge which if removed from the supergraph results in the largest drop in validation accuracy. To bring back the supergraph to convergence a few epochs of further training is carried out between operator selection.

Experiments show that just by carefully thinking about the finalization step in differentiable one-shot NAS, one can obtain much better performance. In fact, one does not even need architecture weights at all! Don't worry about complicated bilevel optimization, gumbel softmax approximation, etc. Just train a supergraph and pick operators progressively.

Comments:

- The paper is wonderfully written! Thanks!

- As I read a paper I try to think without looking at the experiments, what set of experiments I would try to run to prove/disprove the hypotheses proposed. Afterwards I go through the experiments and see if those experiments were actually run (or if they differed why). In this case, every experiment and more were already run. Particularly towards the end I was thinking what if we just got rid of all the alphas and just trained a supergraph as usual and did the PT finalization as proposed. And lo and behold, it actually works better!

- This paper is actually throwing a big wrench in one-shot differentiable NAS literature. Many papers are being written which try to improve/fix DARTS and DARTS-like methods. If I were to believe the experiments, I don't actually need to do any of that. I have some questions I hope to discuss with the authors:

1. Is all the complicated bilevel optimization (often popular as 'metalearning' currently) not useful in the case of NAS? (This is not really the authors' burden to answer but I am just hoping to see if they have any insights.)

2. Can we view the PT finalization step as a progressive pruning step?  So if I were to turn this into a method which produces a pareto-frontier of models (e.g. accuracy vs. memory/flops/latency etc), we first train a big supergraph and then progressively prune out operators one at a time as proposed here and take a snapshot of the supergraph and plot it on the x-y plot (where say x is latency and y is accuracy) and pick the ones clearly on the pareto-frontier and train them from scratch? (Again not really authors' burden but curious if they have any insights)

3. Figure 4 suggests that training the supergraph anymore than 20 epochs only hurts performance (no matter which finalization procedure is used, of course PT has far less of a drop). Does bilevel optimization actually hurt with weight sharing?

---

> ### Author Response · Authors · 2020-11-19
> **Response to AnonReviewer4**
>
>
> Thank you for your positive comments. We respond to your questions below.
>
> *1. "Is all the complicated bilevel optimization (often popular as 'metalearning' currently) not useful in the case of NAS? (This is not really the authors' burden to answer but I am just hoping to see if they have any insights.)"*
>
> In our humble opinion, using advanced methods (e.g., second-order) for solving bilevel optimization in Darts might not be that useful. With the proposed architecture selection method, all that matters is to train a good underlying supernet. And there are many simpler ways to accomplish this goal than formulating NAS as bilevel optimization and trying to solve it. For example, simply fixing $\alpha$ to uniform and training the model weight already works pretty well in many cases, as discussed in the paper.
>
> We also tried more complicated bilevel optimization methods before. Initially, we suspected that the skip connection domination might be related to the ignored hyper gradient direction. But it does not seem to be the case. We tried various second-order methods and used exact hessian-vector-product (hvp), but we found that they have the same issues as Darts (e.g., skip connection domination). Moreover, the computational overhead is very high as the NAS supernet is substantially larger than the ones used in the second-order meta-learning methods.
> \
> \
> *2. "Can we view the PT finalization step as a progressive pruning step? So if I were to turn this into a method which produces a pareto-frontier of models (e.g. accuracy vs. memory/flops/latency etc), [...]"*
>
> Yes, we think it can be viewed as progressive pruning guided by the supernet’s validation accuracy. As you suggested, the selection criterion for every edge can be extended to many multi-objective settings. This can be a good future direction to explore.
> \
> \
> *3. "Figure 4 suggests that training the supergraph anymore than 20 epochs only hurts performance (no matter which finalization procedure is used, of course PT has far less of a drop). Does bilevel optimization actually hurt with weight sharing?"*
>
> We do find it reasonable to assume that there exist better ways to train the continuously relaxed supernet than bilevel optimization. For example, by fixing $\alpha$ as uniform and just training the model weight, we observe improved validation accuracy of the supernet on NAS-Bench-201. Our proposed selection method on this supernet is stable and largely improves the final performance. We add Figure 8 in Appendix A.6 to illustrate this. Therefore, the minor drop in Figure 4 is mainly due to the suboptimal hyperparameter settings of Darts’ supernet training used by NAS-Bench-201. The authors of NAS-Bench-201 simply follow the hyperparameter of Darts when producing the benchmark, even though the supernet and training protocol of NAS-Bench-201 differs a lot from the original Darts’ space [1].
> \
> \
> **References**
> 1. Dong et al. NAS-Bench-201: Extending the Scope of Reproducible Neural Architecture Search. ICLR2020

---

> > ### Comment · AnonReviewer4 · 2020-11-23
> > **Thanks for the response!**
> >
> > I agree with most of the above statements and thanks for the clarification on Figure 4.

---

### Official Review · AnonReviewer1 · 2020-10-29
**Interesting analysis, however the proposed method not that exiting**

**Rating:** 7
**Confidence:** 5

**Review:**

-- Short Summary --

This paper proposes a new policy for selecting the optimal architecture in neural architecture search (NAS), in particular for methods that involve a one-shot model and that deploy gradient-based methods for the search. The proposed algorithm sequentially prunes and fine-tunes the one-shot (aka supernet or weigh-sharing) model until the operations that contribute the most in the one-shot validation performance remain at each edge of the cell (represented as a DAG).

-- Detailed comments --

Positive points:
- The structuring of the sections.
- The paper investigates an interesting aspect of gradient-based NAS algorithms.
- I liked the theoretical justification on the issue with the high \alpha values for skip connections in many DARTS [1] and the experimental backup for that in table 1.

Issues and concerns:
- The proposed algorithm does not seem that elegant and still uses another proxy for true performances of stand-alone architectures, even though indirectly in a lower level, i.e. the drop in performance of the one-shot model after removing one operation at a time in each edge is used to assess the optimal operations operation that will be present in the final architecture. Furthermore, this procedure seems to induce quite some variance by randomly sampling the edges in the cell and then executing the operation drop & fine-tuning steps. Did the authors investigate this by running multiple times their proposed procedure on the same one-shot model?
- Another issue with the algorithms seems to be that it does not scale well with the number of operation and edges in the cell.
- I am not fully convinced by the experiment conducted in 3.1. It seems the authors discretize only one edge of the supernet and fine-tune it further with the discretized edge. In this case the other edges (not discretized) contribute to the network performance together with the single discretized edge and this might be misleading when assessing the importance of an operation. Ultimately, we care about the final network performance, and how this single operation in the discretized edge would perform when combined with all other possible choices in the other edges (still intractable to compute of course). Another potential issue is that the fine-tuned supernet might not correlate well with the discretized stand-alone architecture performance. Can the authors please elaborate on this experiment further in more details?
- Do the theoretical and experimental backup for that hold for other sampling-based methods, e.g. GDAS [2] or SNAS [3]?
- I think the reported search costs in Table 2 are inaccurate. The authors should also add on top of their method the costs of the base algorithm (e.g. 1GPU day for DARTS 2nd order on one GTX 1080Ti).

Minor:
- There are a lot of grammatical errors throughout the text. I would recommend to do a detailed proof read of the paper.
- section 3.2, second paragraph: The acronym VGG does not stand for vanilla networks
- The data in Fig. 1 would look better as a scatter plot, which would also more nicely show the miscorrelation between the 2 quantities that are being compared.
- I like the structuring of the paper, however I think the experiment in the conclusion should not be there, but in the benchmark tables.
- In conclusions: why do you fix \alpha = 0? Did you mean \alpha = 1?

-- References --

[1] Hanxiao Liu, Karen Simonyan, and Yiming Yang. Darts: Differentiable architecture search. In ICLR 2019

[2] Xuanyi Dong and Yi Yang. Searching for a robust neural architecture in four gpu hours. In CVPR 2019

[3] Sirui Xie, Hehui Zheng, Chunxiao Liu, and Liang Lin. SNAS: stochastic neural architecture search. In ICLR 2019

---

> ### Author Response · Authors · 2020-11-19
> **Response to AnonReviewer1 (Part 2)**
>
> **Issues and Concerns (cont)**
>
> *4. "Do the theoretical and experimental backup for that hold for other sampling-based methods, e.g. GDAS [2] or SNAS [3]?"*
>
> We primarily consider methods with continuously relaxed supernets, following previous analytical works [1, 2]. SNAS (when the temperature is annealed to low levels) and GDAS can be considered as discrete sampling-based methods as they sample one single child architecture at every iteration of the search phase. So there is no clear notion of continuous relaxed supernet like Darts. In fact, although SNAS and GDAS are similar to Darts appearance-wise, they behave much closer to another line of NAS methods: single-path one-shot methods [6]. These methods have to deal with another set of issues such as instability of training due to model forgetting [3, 4]. Moreover, our method can improve continuous sampling-based methods such as SDarts [5]. As shown in Table 2, SDarts+PT (2.56% average test error) outperforms SDarts (2.67%).
> \
> \
> *5. "The reported search costs in Table 2 are inaccurate. The authors should also add on top of their method the costs of the base algorithm (e.g. 1GPU day for DARTS 2nd order on one GTX 1080Ti)."*
>
> As mentioned in the response to the second concern in Part 1, we have already included both search time and architecture selection time in Table 2.
> \
> \
> **Minors**
>
> Thank you for your suggestions, we proofread and improved the writing as you suggested. We will continue to refine the paper; For experiments in the discussion section, we initially put them there due to the 8-page limit of the submission. But since we are granted one extra page now, we move the discussions on uniform $\alpha$ to a subsection in Section 6.3.; For fixing $\alpha=0$, setting $\alpha$ as 0 or 1 are both ok. $\alpha$ is the input to the softmax function (section 2 paragraph 1) and can be set to any constant since the softmax function will always map it to a uniform output.
> \
> \
> **References**
> 1. Zela et al. Understanding and Robustifying Differentiable Architecture Search. ICLR2020
> 2. Shu et al. Understanding Architectures Learnt by Cell-based Neural Architecture Search. ICLR2020
> 3. Zhang et al. Overcoming Multi-Model Forgetting in One-Shot NAS with Diversity Maximization. CVPR2020
> 4. Niu et al. Disturbance-immune Weight Sharing for Neural Architecture Search. Arxiv:2003.13089
> 5. Chen et al. Stabilizing Differentiable Architecture Search via Perturbation-based Regularization. ICML2020
> 6. Guo et al. Single Path One-Shot Neural Architecture Search with Uniform Sampling. ECCV2020

---

> > ### Comment · AnonReviewer1 · 2020-11-21
> > **Reply to authors and increasing the score to accept**
> >
> > I thank the authors for their detailed reply and for updating their paper with the proposed suggestions. This paper shows a fundamental flaw in the designing of one-shot NAS algorithms and a simple algorithm on how to circumvent that. I am increasing my score to Accept.
> >
> > Further suggestions (minor):
> > 1. "Darts" --> "DARTS"
> > 2. It would be useful to include the cost analysis somewhere in the paper.

---

> > > ### Author Response · Authors · 2020-11-24
> > > **Thank you for increasing the rating**
> > >
> > > We thank reviewer 1 for raising the score and providing further suggestions. We will incorporate them in the camera-ready version of the paper.

---

> ### Author Response · Authors · 2020-11-19
> **Response to AnonReviewer1 (Part 1)**
>
> Thank you for your detailed comments. Below we respond to your questions and concerns.
>
> **Issues and Concerns**
>
> *1. "The proposed algorithm does not seem that elegant and still uses another proxy for true performances of stand-alone architectures, [...] Furthermore, this procedure seems to induce quite some variance by randomly sampling the edges in the cell and then executing the operation drop & fine-tuning steps. Did the authors investigate this by running multiple times their proposed procedure on the same one-shot model?"*
>
> The effectiveness of the proposed method is evidenced by its consistent improvement on various search spaces (NAS-Bench-201, Darts’ space, and S1-S4), base methods (Darts, SDarts, and SGAS) datasets (e.g., cifar10, cifar100, and svhn).
>
> We would like to emphasize that the numbers reported in Table 2 (marked by “avg”) capture the randomness of both the search and retrain phases; As discussed in the table footnote, we run both search and retrain under multiple seeds and report the mean and std of the test errors. This differs from many previous NAS works (e.g. Darts [1]) as they only run the retrain phase multiple times on the best architecture found (corresponding to the “best” entry of our method in Table 2). As shown in Table 2, the variance of the proposed method is small (below 0.1%). Back to your second point, when running the proposed selection method several times on the same pretrained supernet, the performance of these obtained architectures also stays relatively close to each other, indicating that the proposed algorithm is stable.
> \
> \
> *2. "Another issue with the algorithms seems to be that it does not scale well with the number of operation and edges in the cell."*
>
> According to Algorithm 1, the fine-tuning step happens only after we make an operation decision for a certain edge. Therefore, increasing the number of operations adds negligible cost to the proposed algorithm, since one extra operation only adds one extra forward pass over the validation set to compute the validation accuracy. Besides, the proposed algorithm scales linearly w.r.t. the number of edges. Concretely, let $E$ and $O$ denote the set of edges and operations, and let $T_{tune}$ denotes the cost for fine-tuning and $T_{infer}$ denotes the cost for computing the validation accuracy of the model, then the total cost can be written as $O(|E| T_{tune} + |E||O|T_{infer})$. Since $T_{infer} \ll T_{tune}$, the total cost is approximately $O(|E| T_{tune})$.
>
> The search costs reported in Table 2 have already included the time to train the supernet. Taking Darts+PT as an example, its search cost is “0.8 GPU days = 0.4 GPU days to train the supernet like Darts (1st order) + 0.4 GPU days to perform our algorithm”. So overall it is efficient to run the proposed algorithm.
> \
> \
> *3. "I am not fully convinced by the experiment conducted in 3.1. [...] Another potential issue is that the fine-tuned supernet might not correlate well with the discretized stand-alone architecture performance. Can the authors please elaborate on this experiment further in more details?"*
>
> In Section 3.1 we are considering a simpler setting -- selecting the best operation for just one edge on the supernet. As shown in Figure 1 and Figure 2, the use of alpha is problematic even in this simpler setting, let alone the more complicated and intractable case where the single operation is combined with all possible choices on other edges as you mentioned.
>
> Our primary focus is to investigate Differentiable NAS from the architecture selection perspective, given a pretrained supernet. We think that the second point of this concern raises a general question on the effectiveness of the mixture supernet for the NAS problem, and it applies to most differentiable NAS methods. Since they all assume that the mixture supernet is informative in terms of finding a good architecture. We think that our work in fact paves the way for future research to better answer this question: If we were to evaluate the mixture supernet in terms of representing the strength of its child models, we need to be mindful about how to extract them from such a supernet, as a bad selection method can mislead us. Compared with the previous magnitude-based architecture selection, the proposed method can consistently extract better architectures from the unmodified supernets across base models, search spaces, and datasets. In this sense, our method can be considered as attaining a higher lower bound than the magnitude-based selection on how useful the supernet is for solving the NAS problem, given the circumstance that directly evaluating every sub-network is extremely intractable.

---

### Author Response · Authors · 2020-11-19
**Revision Summary:**

We thank all four reviewers for the detailed reviews and suggestions. We have made the following changes to the paper in our revision, following the comments from our reviewers. All major revisions are colored blue.

* Moving the experiments on Darts+PT (fix $\alpha$) from discussion to section 6.3 (suggested by reviewer 1)
* Extra figures like Figure 1 in Appendix A.5 (suggested by reviewer 2)
* The trajectory of Darts+PT (fix $\alpha$) on NAS-Bench-201 in Appendix A.6 (following comments by reviewer 2)
* Ablation study on the number of fine-tuning epochs in Appendix A.7. (suggested by reviewer 1)
* ImageNet experiment in Appendix A.8 (following comments by reviewer 3)
* Minor fixes (suggested by reviewer 1 and reviewer 2)

We also attached the appendix to the pdf submission for ease of reference (was in the supplementary material folder).

---

### Comment · ~Najmeh_Fayyazifar1 · 2021-01-18
**How architecture evalation is performed?**

Dear authors,

I have difficulties understanding how you have evaluated the operation strength when there are more than 2 operations in the search space. In the case of DARTS search space where there are 8 operations, based on your proposed algorithm, you select an edge randomly, and remove one operation each time from the operation set, then how you evaluate the network accuracy with 7 remaining operations? Do you perform similar to DARTS where they choose the operation with the maximum value of alpha, in this case, the maximum among 7 remaining operations?

Thanks in advance,

---

> ### Author Response · Authors · 2021-01-20
> **Author's response**
>
> Hi Najmeh,
>
> Thank you for your interest in our paper. In Algorithm 1, to evaluate an operation on a selected edge. We remove it from that edge and compute the validation accuracy of the remaining supernet; we do that for each operation on that edge separately will give us a score (i.e. validation accuracy) of each operation. The operation, upon removal, causes the largest drop in validation accuracy (compared with unmodified supernet) will be selected as the final operation on that edge. You can refer to Algorithm box 1/2 or section 4.1 for further details.

---

> > ### Comment · ~Najmeh_Fayyazifar1 · 2021-01-22
> > **How architecture evalation is performed?**
> >
> > Thank you Authors for your reply.
> >
> > I have read your paper quite a few times but it was not very clear to me how the process of computing validation accuracy is done. From your description here, I would conclude that you ignore all alpha values, and for all edges on the graph, you select every possible operation one by one, train the graph from scratch for each operation, and compute validation accuracy. If that's true, then what is the advantage of using darts to compute alpha values?  Isn't it an extensive training/ validation process using all possible combinations of edge connections and operations?
> >
> > One other request, when do you expect to share your code publicly?
> >
> > Thanks again for further clarification,

---

> > > ### Author Response · Authors · 2021-01-23
> > > **Further response**
> > >
> > > Hi Najmeh,
> > >
> > > Alpha values are not used in our architecture selection process; it is no different than any other weights. The operation selection process is done edge by edge so it is not combinatorial. When evaluating operations on a given edge by removal, we compute the validation accuracy of the supernet directly, i.e. the supernet is not trained from scratch.
> > >
> > > We are working on extracting the code from our large codebase; it should be out in roughly 3 weeks.

---

### Author Response · Authors · 2021-04-12
**Code Repo**

Hi everyone,

The code for this paper can be found at https://github.com/ruocwang/darts-pt (not indexed by the Google search engine for some reason).

---

### Decision · Program_Chairs · 2021-01-07
**Final Decision**

**Decision:**

Accept (Oral)

**Comment:**

This paper proposes a new selection paradigm for selecting the optimal architecture in neural architecture search (NAS), in particular for methods that involve a one-shot model and that deploy gradient-based methods for the search. Basically, the paper focuses on examining the max selection very closely and found the magnitude of architecture weights are misleading. Instead, the paper proposes much more intuitive finalization step, pick the operator that has the largest drop in validation if the edge is removed. All reviewers agreed that the idea is interesting, the paper is well-written, and the results found in the paper are interesting. In addition, author response satisfactorily addressed most of the points raised by the reviewers, and most of them increased their original score. Therefore, I recommend acceptance.